# Functional Implications of the Prosomeric Brain Model

**DOI:** 10.3390/biom14030331

**Published:** 2024-03-11

**Authors:** Luis Puelles

**Affiliations:** Department Human Anatomy and Psychobiology and IMIB-Arrixaca (Murcia Institute for Biomedical Research), University of Murcia, LAIB Building, Avenida Buenavista s/n, El Palmar, 30120 Murcia, Spain; puelles@um.es; Tel.: +34-600887906

**Keywords:** brain models, columnar model, neuromeric models-synthetic model, prosomeric model, brain functions, modularity

## Abstract

Brain models present a viewpoint on the fundamental structural components of the brain and their mutual organization, generally relative to a particular concept of the brain axis. A model may be based on adult brain structure or on developmental morphogenetic aspects. Brain models usually have functional implications, depending on which functional properties derive from the postulated organization. This essay examines the present scenario about brain models, emphasizing the contrast between columnar or other longitudinal models and transverse subdivisional neuromeric models. In each case, the main functional implications and apparent problems are explored and commented. Particular attention is given to the modern molecularly based ‘prosomeric model’, which postulates a set of 20 transverse prosomeres as the developmental units that serve to construct all the cerebral parts and the particular typology of many different neuronal populations within the forebrain and the hindbrain, plus a number of additional spinal cord units. These metameric developmental units (serially repeated, but with unique molecular profiles) confer to this model remarkable functional properties based mainly on its multiplicity and modularity. Many important brain functions can be decomposed into subfunctions attended to by combined sets of neuronal elements derived from different neuromeres. Each neuromere may participate in multiple functions. Most aspects related to creation of precise order in neural connections (axonal navigation and synaptogenesis) and function is due to the influence of neuromeric anteroposterior and dorsoventral positional information. Research on neuromeric functionality aspects is increasing significantly in recent times.

## 1. Introduction

The brain models I’ll refer to are the anatomic ones, which may or not have a developmental basis; the prosomeric model is eminently developmental, being based on embryonic transverse neuromeric units as fundamental constituents of the brain. Morphologic models of the brain postulate a particular viewpoint on its basic structural constitution. This implies giving both an *ontology* of its fundamental parts, and a general description of the *mutual arrangements* of these parts usually represented in a schema. In the case of the brain, which is an elongated structure, a length *axis* orienting our positional descriptions and section planes needs to be *defined* and *justified.* Any arbitrary line traced along the schema, or particularly through the ventricular fluid filled cavity, is not sufficiently realistic; the axis is a fundamental morphologic causal concept that needs to be based on solid developmental processes. The prosomeric model fundaments its length axis along the median floorplate, induced early on by the notochord (a process emerged evolutionarily at the origin of vertebrates; see Figure 1 and Figure 2C).

In any case, human curiosity tends not to be satisfied with a mere schema of parts. If you were given a detailed list of the parts of a clockwork mechanism, accompanied by abundant schemata of how each mechanical part relates wheel to wheel to the others, you probably would wonder about the general function of the whole mechanism, or the specific functions of each subset of parts. A morphological brain model similarly is not an end by itself, as a schematic map could be, but often implies predictions or expectations about the function of the various interactive parts. Here developmental models refer to developmental processes and adult fates, whose adult consequences refer to functions. The functional question thus emerges in neurobiology, particularly when novel brain structural models are proposed: Which are the novel developmental and/or functional implications of this brain model you want me to work with? Does it offer deeper or more abundant insights than preceding models? It is expected of any minimally interesting new model that it demonstrates both a strong underpinning of its structural details by convincing morphological and developmental evidence (establishing credibility of the model itself), and novel functional implications of some importance (interest of the model).

Surprisingly, many neurobiologists or neuroclinicians are not much aware of which neuromorphological model they are using in their professional work. Even when they have a notion about it, their comprehension of the relevant assumptions and implications may be quite foggy. Some experts actually think, wrongly, that they are not using any specific brain model (they think they just deal with known *facts* about the brain). It turns out that *everybody necessarily uses a model*, knowingly or not, since otherwise we cannot deal with the complexity of the brain, nor understand its nomenclature. Moreover, most professionals also do not know precisely how many different brain models exist that may be mutually compared, in order to choose among them for optimal guidance in personal professional use or for teaching purposes. This mental hole partly occurs because professionals only are exposed superficially to a brain model in the very first hours of their long past studies of neuroanatomy, and they never saw afterwards a deep discussion on that subject.

Indeed, textbooks of neuroanatomy and neuroscience treatises invariably *use a specific brain model* chosen among a variety of them available, but habitually do not present it openly as an optional construct, identified under the name of ‘model this or that’. Such a presentation would require explaining at least its basic assumptions and main underpinning data, thus opening it to criticism and contrary evidence, a desirable, but hardly practiced option. This sort of open and non-dogmatic presentation rarely occurs in practice. The usually unnamed model followed by the book authors is instead introduced as a dogmatic definition briefly telling the reader that the brain is made of a number of parts x, y, z. The briefness of the approach apparently intends to orient the reader towards placing his/her attention on other more important aspects of the book’s message. The major aim lies in giving a minimum of conventional names needed for reading further about these brain parts, disdaining consideration of their structural arrangements, axial relationships, and possible functions. Sometimes the brain length axis postulated in the book’s model is mentioned or illustrated, but is never discussed causally or comparatively. There is no analysis comparing one with another the existing alternative brain models, which are not even mentioned. The reader is thus attracted subliminally to having faith in the particular model used by the book author, without being offered a corresponding rationale (as the author probably was in his own time). The message is therefore unwittingly transmitted across generations that the presented basic structural knowledge is not a target for attention, since its core items have apparently been repeatedly corroborated by all major experts over many years, so that they must represent a widely accepted consensus on which you can depend without further thought. These defective and irrational introductions to neuromorphology are more of a habit or tradition than a consciously selected approach, since nobody intends to fix an eternal dogma. Simply, old ideas tend to become fixed as if they are facts, which only need to be briefly stated; no arguments about underpinning evidence are needed (like in: “we all know that the Earth is flat.”). The existence of such defective introductions of brain models represents an easily verifiable fact; just check the beginning of any major neurobiological source you may have around to identify the dogmatic nature of the unnamed structural brain model given therein.

The human brain happens to be *the most complex living structural system composed of interactive parts we know about* on Earth. The neuroanatomic source literature on brain models is manifestly opaque and seems therefore oddly unhelpful from a didactic viewpoint. Neurophysiology and neuropsychology are derived neurobiological sciences that study specifically brain functional phenomena at different levels of complexity, but they traditionally attend even less to brain structural models than the neuroanatomic mother science.

There are possibly two additional explanations of this traditional lack of epistemological attention to brain models. One possibility is that, historically, humanity needed several millennia in order to achieve a sufficiently complete and reliable list of the diversity of anatomic parts present in the human brain, largely because the initial methods of analysis were too crude (see historical works such as [6,7] to get a glimpse on that aspect). The intrinsic insecurity felt in available anatomic knowledge did not make easy having a consensus on which are the most meaningful primary structural and functional units of brains in general. Note we may distinguish three successive epochs of brain knowledge up till the present state. These epochs were dominated successively by, first, anatomic dissection and naked-eye analysis, second, microscopic neurohistology and neurophysiology and, third, molecular biology plus neuropharmacology.

A recent expanded ontology of the developing mouse brain assessing all its named anatomic compartments irrespective of size (distinct neuronal brain subregions) was elaborated by the present author for the Allen Developing Mouse Brain Atlas during 2008–2013. It was found that over 2500 differently named domains apparently exist per each brain halve, so that there are roughly 5000 such parts functioning together in the developing mouse brain (developingmouse.brain-map.org (accessed on 5 March 2024); see Supplementary Data, the Ontology archived on March 2010). This schema of parts is thought to be shared by most mammals including humans, though the latter apparently generate more subdivisions at given places like the cerebral cortex. At the end of 2023 the Allen Institute presented a novel mouse brain map generated with single-cell transcriptomic molecular methodology (based on analysis of which genes are differentially active in specific neuronal populations, depending on their localization). This map is called The Whole Mouse Brain Atlas. It apparently distinguishes some 5300 different cell populations in each brain half, without entering into the many non-described cortical areal subdivisions, which might lift the number at least to 5500, thus bringing the approximate total of the brain to 11,000 molecularly distinguishable parts (doubling in 2023 our estimate of 2013). It is obviously not practical to generate a realistic and easily manipulable or explorable brain half model with 5500 parts.

Introductory texts of neuroanatomy clearly do not approximate at all this presently existing level of knowledge, giving maximally only a few dozen anatomic names for the basic orientation of readers. Due to their mostly ancient roots in research from the 19th century (200 years ago), a good number of these basic, widely established concepts are not even strictly consistent with the novel molecular knowledge. Anatomic brain parts identified some hundreds of years ago are crude approximations to reality as we see it today. This historical contrast existing between textbooks and current journal literature reveals that most present-day professional neurobiologists are handicapped as regards appraising the full structural and functional complexity now described in the mammalian brain, needing increasingly the help of computers. Active researchers normally specialize in particular subregions of the brain, as a way to deal with such complexity, and thus are likely to be expert only on given functional subsystems. They consequently tend to pay a price in their diminished perspective on the overall scenario and thus normally do not control a whole brain model. I know this from personal experience. It took me many years of exploration and reading of the sources to be able to visualize mentally in sufficient detail the whole brain, and this only thanks to the help of a sound model to be presented below under the Section 2.6. Before, we will examine some other antecedent models (Section 2.1, Section 2.2, Section 2.3, Section 2.4 and Section 2.5)

## 2. Brain Models

### 2.1. The Columnar Model

An additional explanation of the described poor cognitive scenario on brain models is conceivable. It relates to the fact that neuroscientists, book authors, and brain atlas cartographers have basically been using the same brain model–known to experts as ‘the columnar model’- for the last 100 years or so. This model was originally proposed by C. J. Herrick in 1910 on the background of important comparative studies performed around the end of the 19th century on the sorts of functionally different fiber constituents of the spinal and cranial nerves and their selective connections with anatomically and functionally distinct sets of neurons found in the brainstem and spinal cord of vertebrates [8,9,10]. These then newly identified source or target neuronal groups were described as forming longitudinally disposed (and dorsoventrally stacked) sensory and motor neuronal *columns*, which were held to represent the long sought fundamental structural parts of brain functional organization (thus ‘the columnar model’ name; see Figure 2A [11]). Moreover, Herrick extrapolated speculatively this columnar structure notion also to then relatively unexplored forebrain parts, namely the diencephalon and the telencephalon (note that both regions are devoid of nerves providing sensorimotor in- or outputs, if we except olfactory input; see Figure 2A).
Figure 2Diagrams of three brain models, one columnar (**A**) and two neuromeric (**B**,**C**), modified from Figure 5 in [12] (with permission). (**A**). Schema original from Herrick [11], showing his initial columnar model of the brain. This model postulates an extension of the four brainstem functional columns into the forebrain (colored fields in diencephalon and telencephalon). Towards that aim, the diencephalic columns were defined to be ‘longitudinal’, irrespective of their topography relative to the brain’s axial incurvation (see cephalic flexure under the mesencephalon; compare with cf in (**C**)). Note Kuhlenbeck [13,14] later showed that neither epithalamus nor dorsal thalamus (yellow and violet domains) are continuous with the telencephalon, since they are separated by parts of the roof plate (compare (**B**,**C**); particularly the position of the anterior commissure [ac]). (**B**). Diagram of the synthetic neuromeric model, original from Bergquist and Kallén [15]. These authors held that the four longitudinal zones of His (floor, basal, alar, and roof) are crisscrossed orthogonally by the transverse boundaries of forebrain, midbrain and hindbrain neuromeres (numbered by them as 1–13), thus defining a number of quadrangular migration areas. Note their model admitted that some longitudinal zones are not present along the whole length of the brain. The rostral ends of the floor and roof plates remained undefined, though the series of neuromeres along the bent axis (red dash line) imply that the rostral end of the axis lies in the hypothalamic neighborhood of the optic chiasma (visible as a thickening next to the rostral end of His’ sulcus limitans or alar-basal boundary). Ulterior studies in the hindbrain led to accept an increase in the number of rhombomeres (compare with (**C**)), given that a number of them are cryptic (i.e., non-visible as bulges morphologically, though they are detectable with molecular methods and experimental fate maps). (**C**). Diagram of the prosomeric neuromeric model of Puelles and Rubenstein, original from the author, as published with minor variations in [16,17,18,19,20]; compare with Figure 1. A global similarity is evident with the antecedent synthetic neuromeric model (compare (**B**)). However, the floor plate (fp; blue) is now defined molecularly and experimentally (it is induced by the notochord) as ending rostrally at the mamillary body (now understood to be a rostral entity). The roof plate (rp, yellow) is defined by experimental fate mapping as ending at the anterior commissure (ac) above the preoptic area (poa/ap). The red axial dash-line, similar to that in B, is the molecularly defined (*Shh* and *Nkx2.2* markers) alar-basal boundary (a/b), not being referred expressly to a ventricular sulcus. The rostralmost prosomere is that identified as hp2 (compare Figure 1), and its green-marked rostral limit extending between floor and roof plates is the acroterminal rostralmost domain of the brain (AT), where the rostral ends of the parallel floor (fp), basal (bp), alar (ap) and roof (rp) plates lie, and where the paired lateral walls of the tube (bp, ap) are continuous left to right. The telencephalon is interpreted as a bilateral dorsal alar evagination from the hypothalamic hp2 prosomere (i.e., each hemisphere is independent from the roof plate), though the rostral part of the prethalamic diencephalic prosomere is also partly evaginated (not shown; imagine a rostrodorsal red flap evaginated into the neighboring telencephalic vesicle).
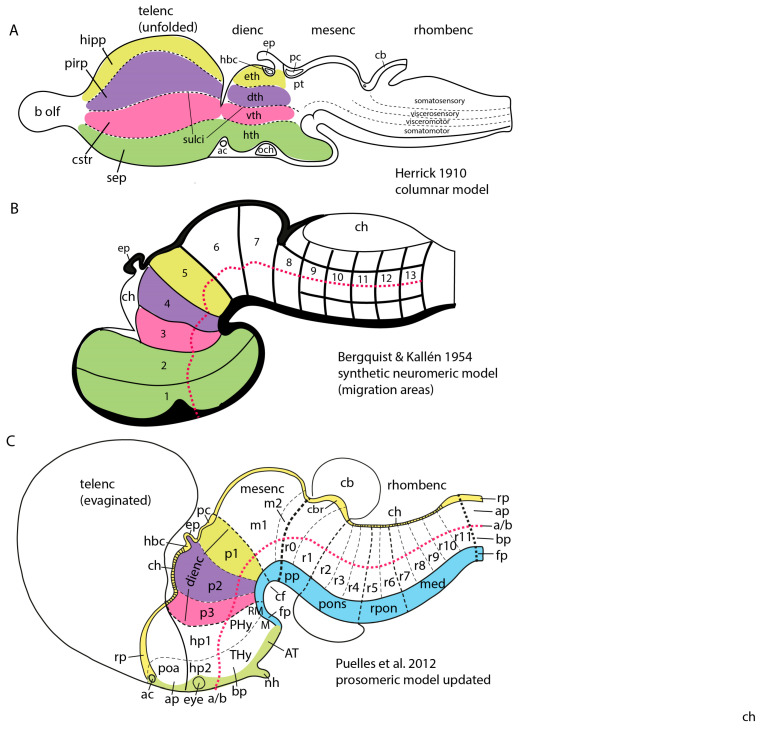



The 1910 columnar model in essence postulated that the whole brain was made of a few dozens of neuronal columns distributed over a few anatomic regions (forebrain, midbrain, hindbrain, and spinal cord), where each unitary column would embody a different sensory, motor, or sympathetic/parasympathetic function. The primary sensory columnar units would intercommunicate with a hierarchy of integrative higher-level centers in order to knit sensory input and analysis into neural representational models of the body and the world, whose computations were held to serve for planning efferent motor output to produce reflex or voluntary behavior. So-called ‘higher’ mental functions were thought to emerge collaterally with increased evolutionary encephalization (disproportionate growth of particular higher brain parts such as the basal ganglia and the cerebral cortex, or the cerebellum, relative to the animal’s body weight; Figure 2A). As a consequence, more complex networked aspects of internal connectivity are acquired during brain evolution. Herrick was thus an important neurobiologist that gave cogent attention to the functional implications of his brain model in his research articles (usually examining in great detail simple amphibian brains such as newts and salamanders), or in his numerous functional divulgation articles published throughout his carrier. He wrote an impressive list of books ([21,22,23,24,25,26,27]; see the large list of his functional publications in the last reference). Somewhat surprisingly for a neuroanatomist, Herrick argued vehemently that ‘neuromorphology should be functional’; for instance, he openly discrepated with H. Kuhlenbeck [13,14], a German comparative neuroanatomist and embryologist who likewise defended the ‘columnar model’ after studying the major lineages of vertebrate embryos on the basis of ‘strictly formanalytic’ developmental analysis (i.e., embryologic structural studies devoid of functional applications). Herrick clearly preferred the parallel *morphofunctional* studies on amphibians of his friend Coghill (whose biography he wrote and commented in detail; [25,26,27]). A similar primarily functional outlook was demonstrated in a then revolutionary book where Herrick’s colleague J. B. Johnston [28] presented the early columnar functional theses.

After the first and second world wars, the American school of neuroanatomy grew considerably around Herrick’s columnar model and the latter gained much scientific ascendancy and became implicitly or explicitly accepted by members of European or Asiatic neuroanatomy schools. This widespread support eventually led by the end of the twentieth century to the gradual crystallization of the ‘functional’ columnar model assumptions as representing undoubted facts, rather than remaining tentative theoretical constructs, as they were at the beginning (and in fact continued being 100 years afterwards). This worldwide scenario probably was the immediate cause of the cursory and dogmatic treatment given to this model in most source books presently in use, and thus possibly also explains why most present-day neurobiologists ignore not only that they are using the original columnar model of 1910 (or one of its subsequent minor variants; see e.g., [29]) in their professional activities and teaching, but also ignore whether the columnar modelling of the brain has turned out to be satisfactory in the long run or not. 

Modern physiologists only partly describe their results in terms of the classic columns (mainly in relation to brainstem sensory systems), and physiology books normally leave out treating parts of the brain whose connections and functions remain hardly understood under this model. One may point out as Cinderella-like brain places which are hardly mentioned in functional perspective: the preoptic area, many hypothalamic subregions, the prethalamus, the pretectum, the prepontine hindbrain, as well as large parts of the telencephalon. After more than 100 years of research with progressively more potent methodologies, these brain areas have not been explained functionally by the widespread columnar modelling approach, and some of the apparently available partial explanations are subject to recent critical comments (see [30] on the visual system). 

Over the years some minor variants of the columnar model were published, often without discussing in detail their differences or similarities with Herrick’s original proposal [11]. The most recent variant of this model by Swanson [29], a neuroanatomist also heavily interested in functional interpretations besides publishing substantial anatomic studies. In general, columnar-based models are characterized by arbitrary (causally unexplained) postulation of a forebrain axis that ends in the telencephalon (though we do not know experimental, developmental or molecular reasons for this assumption, rather the contrary; [18]). The natural incurvations of the embryonic neural tube, notably the cephalic flexure, are largely not taken into account to determine what is longitudinal, that is, columnar (Figure 2A). The adult columns are officially delimited by ventricular sulci *said* to run parallel to the nondescript axis, though these variable landmarks are not easily visualized, particularly in embryos. The columns are thus *assumed* to be delimited by supposedly longitudinal sulci, though various authors have stressed that these -as mapped- sometimes seem to run orthogonal to the bent natural axis (see Herrick images in the Chick Brain Atlas text [31]). The brain’s length dimension and the ‘longitudinal’ nature of its *forebrain* columns is thus a very vague, arbitrary concept in this model. In contrast, the longitudinal nature of the *brainstem* columns is not problematic, representing the most valid contribution of the columnar approach. Even in this case, the developmental formation of the brainstem columns has been qualified substantially by the modern experimental and molecular rediscovery of neuromeres (transverse, rather than longitudinal, structural units, which cause columnar subdivisions; [2]).

The fundamental practical problem of the columnar model is that its columnar units represent late histogenetic phenomena defined as originating at late embryonic stages (after the nerve connections are established). As a result, columnar interpretations leave aside practically all developmental considerations on the early neural tube (patterning, differentiation, cell migrations). The columnar theory, designed by Herrick for providing adult functional implications, lacks any factual underpinning dealing with *causal processes* active during early developmental stages (e.g., molecular positional patterning, proliferation patterns, early bulging of vesicles or neuromeres, neurogenetic patterns, cell migration, areal regionalization, stratification and layering of neurons, axonal navigation, synaptogenesis). This rather outdated model thus essentially cannot deal with how the brain is patterned (regionalized) and constructed as regards acquiring its structural and functional complexity. Importantly, *it deals with too few brain parts,* some of them badly defined. The columnar model simply does not contemplate at all *how* a supposedly initially homogeneous column is transformed into a non-homogeneous adult column (or which are the functional implications of a thoroughly compartmented column such as the modern hypothalamus, for instance, or the multinuclear thalamus; Figure 2A). Given that columnar model-based theory stops at the level of the entire column and its possible overall functional relationship with sensory-motor or sympathetic-parasympathetic peripheral connections in the body, it traditionally does not admit interpretations based on the finer constituent histogenetic phenomena occurring inside the columns. Note there are only a few columns and we already know about at least 5000 different subregional domains in the mouse brain, and about 11,000 molecularly distinguishable neuronal cell types, as stated above. The columnar model is therefore extremely simplistic as regards both subjacent developmental causal processes and functional parts. This implies that the functional implications deduced by use of the columnar model also must be simplistic (for instance, the widespread notion that basal ganglia are merely motor centers is simplistic, as is the idea that the cerebral cortex is an integrative organ). In particular, given that all postulated columns are implied to be either sensory, motor or autonomic regulatory (sympathetic or parasympathetic control of peripheral autonomously functioning ganglia; Figure 2A), it surely must be doubted that all functions performed by the brain belong necessarily to these categories (e.g., compare with evaluation of experience in the amygdala and limbic cortical system, or the largely unknown function of the septum).

In the present scenario we have the problem that the outdated and excessively simplistic columnar model still continues to be used dogmatically in introductory texts (without being acknowledged by name), but it has ceased to be scientifically useful since at least some 50 years now, particularly with regard to the numerous molecular, causal, and complex functional data accrued in this period and the modern appreciation that by the number of distinctly populated parts, the brain is much more complex than was appreciated by Herrick in 1910. The columnar model therefore has become by its widespread subconscious implantation a cryptic mental superstructure blocking perception of more complex patterns. It is cryptic because most modern users of this model are not aware of its existence, nor of its outdated assumptions. Scientists that take this model as providing *realistic knowledge* about the brain are often blinded to the fine phenomena of brain construction and function now studied at molecular-level, *which are not organized in a columnar fashion*. The era of experimental, molecular, and genetic embryologic approaches, which is now some 50 years old, has made it clear that models of the brain are no longer particularly meaningful or useful if they are not based on *developmental molecular causal phenomena*. They must allow us to understand from the beginning how the brain is constructed during development (this construction can go wrong in many directions, sometimes giving rise to subtle malformative consequences having functional impact, such as the Down or Asperger syndromes).

### 2.2. The Main Developmental Models Available

The oldest brain models recorded in the literature reflect conclusions of technically imperfect anatomic dissection of adult brains, either in animals or in the human. They were consequently very tentative when not speculative as regards the brain parts thought to exist and their relationships, or the potential functions attached to them. For instance, until experimental anatomic and physiological study of connections in the brain was possible, it was frequent to postulate what later was discovered to be the wrong direction of signal transmission in many brain tracts (e.g., false ascending signals in the pyramidal tract towards the cortex and/or the corpus callosum, or efferent signals from the brain into the eye through the optic tracts). The plurimillenial long first historical epoch based largely on anatomic dissections led to a second epoch of clarifying neurohistological microscopic work which exploded consequent to the development of the cellular theory and the advance of microscopy and staining methods. This line of analysis resolved a number of classic problems and was soon complemented by comparative neurobiology, experimental study of brain connections, neurophysiology, and neuroembryology (first descriptive and then experimental). Arrival of Darwin’s theory of evolution in 1859, with subsequent developments in genetics, biochemistry and molecular biology, allowed to gradually come to understand that evolutive steps are produced by subtle genomic changes translated by embryos into functioning brain forms via molecularly regulated mechanisms of development. Our understanding of highly evolved brains such as the human one thus depends significantly on our knowledge of how it develops and its genetic background.

Embryological studies performed along the second half of the 19th century produced at least two developmental brain models, which can be seen as having introduced complementary concepts that later were integrated into a synthetic concept: one of the initial notions was the ‘neuromeric model’, which considered *transversal developmental units* called ‘neuromeres’, divided according to the major ‘vesicles’ where they appeared as *prosomeres* (forebrain), *mesomeres* (midbrain), *rhombomeres* (hindbrain) and *myelomeres* (spinal cord) (see Figure 1 and Figure 2B). The other notion was represented by the ‘longitudinal dorsoventral zonal model’, also known as ‘His’s model’, which considered *longitudinal developmental units, called* ‘*plates*’ or ‘*zones*’ (the floor, basal, alar, and roof plates of His [32,33,34]; Figure 2B,C).

### 2.3. The Early Neuromeric Model

The initial ‘neuromeric model’ centered attention on the early existence of series of transverse outpouchings (vesicles) in the lateral wall of the embryonic neural tube, which were delimited by transverse constrictions (producing inner ridges in the ventricular relief). These pouches generate an apparent ‘serial segmentation’ of the neural primordium, organized orthogonally to its length axis. Some of these outpouchings in the brainstem and all of them in the spinal cord were clearly related to the root of given cranial or spinal nerves. These neural outpouchings were eventually christened ‘neuromeres’ [35] (note the suffix ‘mere’ originates from ancient Greek μέρος (méros), meaning “part”; its use in neurobiology implies a ‘repeated partition’). It relates to the descriptor ‘metameric’, which refers to *serially repeated units or partitions* (it is understood that such repeated units each show an *identical basic structural organization*, like the shared floor, basal, alar and roof plates in the case of neuromeres; Figure 2B,C). Neuromeres are thus metameric brain units, likewise as somites, branchial arches, vertebrae, ribs, etc.…are metameric body units). Consistent numbering of the neuromeres was a problem for a time, due to technical problems (irregular embryonic series and delayed fixation, which introduced postmortem fixation artefacts). However, these problems were gradually resolved, leading to an important general review ([36,37]; Ziehen wrote the missing chapter on mammals after the death of von Kupffer). This work covered what seemed comparable neuromeric phenomena in all vertebrate lineages. This crucial work written in German was never translated into English, which limited considerably its ulterior diffusion. Note it was contemporaneous with the antagonic columnar model of Herrick [11].

According to the neuromeric model, the neural tube registers at early embryonic stages a serial transversal vesiculation along its axial dimension (a rostrocaudal pattern), establishing a number of repeated metameric units called neuromeres, out of which the adult brain is built, with cryptic maintenance of the early interneuromeric boundaries. In the molecular era (after the nineteen eighties) this concept was enriched by recognition of the fact that each neuromere is patterned differentially to have a singular molecular profile, that is, to express a partly shared but in fact unique combinatorial set of developmental genes. This starting point determines subsequently after various molecularly guided regionalization and differentiation steps *the particular fate of each neuromere within the brain*. There are 20 cerebral neuromeres (Figure 1 and Figure 2C), plus variable numbers of spinal neuromeres in different species (for instance, there are many more spinal neuromeres in snakes than in frogs; there are roughly as many spinal myelomeres as spinal nerves, according to recent counts).

Note the old authors usually counted less cerebral neuromeres than we do nowadays, because they only counted those showing well delimited bulges. Modernly, a number of *cryptic neuromeres* were identified by gene mapping and experimental methods [2,38,39]; these cryptic units do not bulge out, but are still comparable with standard bulging units in their molecular and developmental properties. The occasional bulging seems to be due essentially to an eventual disproportion between maximal central proliferation versus slower proliferation at the constricted interneuromeric limits. This is however a local and variable *consequence* of neuromery itself and particular molecular profiles, not being strictly needed for the formation of a neuromere (molecular specification and ulterior growth). Modernly we learned that what establishes the neuromeres and their differential fates is strictly their unique molecular profile, irrespective of the overall formt. 

Serial neuromery formed initially part of an ampler theory on head segmentation phenomena (considering also so-called head or cranial vertebrae, metameric cranial nerves, blood vessels, and somites, and metameric branchial arches). This theory reached its acme of interest at the end of the nineteenth century but was gradually left aside afterwards, jointly with the ‘early neuromeric brain model’, roughly coinciding with Herrick’s 1910 publication of his ‘columnar model’ (see however a recent report on blood vessels and neuromeres; [40]). 

A criticism addressed initially to the neuromeric model was that the neuromeres were not known to have any functional implications. Moreover, their visibility as transverse bulges of the brain primordium diminished with time, so that they seemed to become effaced as the neural wall differentiated and became functional (that is, when the longitudinal columns formed instead). The differentiating columns were thus concluded to overlie and efface the neuromeres (a doubtful notion; the supposed mechanism of deletion was never demonstrated). This notion actually turned out to be false, since we can now visualize perfectly neuromere-derived domains in the adult mouse brain with transgenic labeling methods (e.g., Figure 3 and Figure 4). However, at the time (first half of the 20th century), no adult functional consequences seemed to relate to the early neuromeric repetitive units. The subsequent ascent in popularity of the columnar model led to practical disappearance of the notion of ‘neuromere’ in the literature. Only a few occasional authors continued to report neuromeric descriptions; see recent review and reappraisal of modern neuromeric literature in [41].

### 2.4. His’s Model

A second developmental brain model was conceived by the Swiss neuroembryologist Wilhelm His. He discovered the early subdivision of the developing human brain into primary longitudinal ‘floor, basal, alar, and roof plates’, each extending in parallel along the whole neural tube, and adapting to its morphogenetic axial bends [33,34,35]. This is usually known as ‘His’s model’, but it might also be described as the ‘dorsoventral longitudinal zonal model’, since the four longitudinal plates of His are stacked dorsoventrally (roof to floor; Figure 2B,C). This model ascribed neurogenesis (neuronal production) mainly to the basal (ventral) and alar (dorsal) plates. W.His noted that the precociously differentiating basal plates thicken out early on, forming a longitudinal bulge in the ventricular (inner) relief of the brain wall. This bulge is limited by a longitudinal *alar-basal sulcus*, known as the *sulcus limitans of His*. The latter can be followed across the neural tube flexures to a rostral end in the neighborhood of (over or under) the optic chiasma. This implies an axial end in the hypothalamus, rather than in the telencephalon, as was later held arbitrarily by the columnar model; Herrick never discussed the contradiction between his arbitrary forebrain length axis and the developmentally well-supported and generally accepted one of His. These longitudinal plates of His agreed with the underlying notochord, which was later shown to induce primarily the floor plate (and indirectly, interactively with roof plate signals, also places the other parallel longitudinal zones).The basal plate (containing the motor columns) essentially produces neurons involved functionally in motor mechanisms (motoneurons and associated interneurons plus reticular motor elements), whereas the later born neurons of the alar plate (forming the sensory neuronal columns) receive all primary sensory nerve inputs and thus are functionally related to initial sensory analysis. This functional characterization of the primary longitudinal plates of His was not far removed from the notion of alar-basal sensorimotor reflex circuits conceived by physiologists (e.g., [43]).

The longitudinal plates of His reflected early differential fate specification (patterning) of the basal and alar plate derivatives (as well as of the floor and roof ones). It was relatively consistent with the functional notions of the columnar model for the brainstem and spinal cord, since the columnar authors merely divided the alar plate into viscerosensitive and somatosensitive columns, and the basal plate similarly into visceromotor and somatomotor columns (Figure 2A). However, His’s model discrepated with columnar axial notions in its postulated (and demonstrated) rostral end of the longitudinal zones in the hypothalamus (the telencephalon was seen in His’s model as a secondary dorsal outgrowth of the hypothalamus (Figure 2B,C); see recent review in [44]). This implied different morphologic, causal and functional interpretations of the forebrain areas (not so for the brainstem). His’s model did survive in embryologic and anatomic textbooks even after the ascent of the columnar model, insofar as its functional implications were consistent in the brainstem with those of the columnar model. However, the authors following the predominant columnar school usually ignored His’s model as regards its different forebrain axial assumptions (they negated or simply disregarded an alar/basal subdivision of the forebrain; Swanson [29] recently redefined that aspect to extend basal and alar domains in the telencephalon, but tendentiously, because there is no developmental mechanism that can induce floor and basal domains in the telencephalon). In any case, the solid developmental basis of His’s model was the only one available (the columnar model having no embryology at the beginning; Kuhlenbeck [13,14] provided that missing aspect later, also in a somewhat opinionated or preconceived way). His’s model of longitudinal zones finally resulted supported by modern gene expression data and experimental patterning studies, once these arrived in the late nineteen eighties and nineties.

The major problem of His’s subdivision of the dorsoventral dimension of the neural tube into four longitudinal zonal units is that his model in principle does not contemplate the embryonic neuromeric phenomena that serially divide the anteroposterior axial dimension of the whole neural tube. His [32,33,34] did postulate a few roughly defined anteroposterior transverse subdivisions, but these only agree in one neuromere -the isthmic one- with the modern list of 20 neuromeres. As a result of this weak point, His’s model similarly as the columnar model is incapable of explaining the fine subsequent regionalization phenomena occurring within the longitudinal zones or plates due to early neuromeric transverse partition phenomena. These also turned out to be strongly underpinned by emerging molecular and experimental data. These transverse units have thus reappeared importantly in the literature during the current molecular/genetic epoch of neuroscience (these last 50 years or so; see Figure 1 and Figure 2B,C).

### 2.5. The Synthetic School of Neuromery

An alternative developmental brain model unifying purposefully both the transverse neuromeric [36] and the longitudinal zonal [32,33,34] approaches of the late 19th century emerged during the early 20th century in the hands of the Holmgren school [15,45,46,47,48,49,50]. The first three cited authors (P, R, B) were pupils of the Swedish neurohistologist Niels Holmgren, who apparently developed the logical notion of a pattern of checkerboard-like areas of the brain wall that are sharply delimited in the neural tube wall by *the intersection of the series of transverse neuromeric limits with the longitudinal zonal boundaries of His’s dorsoventral zones (they used also His’s ‘natural’ longitudinal axis)*. The other authors were independent followers of that school in Sweden (Källén), Denmark (Vaage) and Holland (Keyser; Gribnau and Geijsberts). I also became a geographically distant follower of this school since the late seventies [51]; my only personal contact was with Kallén, whom I met during a memorable 2002 weekend in Lund (Sweden). I also briefly met Keyser in 2017 in Holland, during a birthday-feast of a colleague. I had studied thoroughly his thesis work on the development of the hamster diencephalon. According to what I call here the ‘synthetic neuromeric model’ each neuromeric unit would be divided dorsoventrally into floor, basal, alar, and roof quadrangular histogenetic areas (Figure 2B). Such intersection areas, studied by these authors with cytoarchitectonic methods and occasional histochemistry (I used whole-mount AChE; see [4]), were modernly corroborated with molecular markers and experimental fate mapping methods, giving rise to the updated molecularly based ‘prosomeric model’ of Puelles and Rubenstein; see next Section ([18,52,53]; Figure 1 and Figure 2C).

The quandrangular neuroepithelial domains were first identified as ‘*migration areas*’ [15], referring to their central maxima of mitotic cells and major central production and radial migration of newborn neurons (Figure 2B). Later, they were described alternatively as ‘*radial histogenetic areas*’ [51], a term emphasizing the overall radial organization of its collection of ventricular progenitor cells plus the derived subset of neurons and glia cells born within each area (see [12]). More recently a third name was proposed, that of ‘*fundamental morphological units*’ (FMUs of [12,19]), thus identifying them as the fundamental (primary) developmental brain components, whose summed up individual histogenesis and morphogenesis jointly produces the complex structure of a particular anteroposterior and dorsoventral block of the mature brain wall. 20 cerebral neuromeres divided each into 4 FMUs (corresponding to their shared floor, basal, alar, and roof domains) makes a total of 80 FMUs as the basic building blocks of the brain (Figure 2C). It is clear that in order to reach 5000 parts in the adult, several rounds of further regionalization of these 80 FMUs need to be considered (secondary, tertiary, … and *terminal microzonal* FMUs of [12]). Although still modest in its pattern of subdivisions, a model encompassing the 80 basic FMU parts may be understood and handled profitably by students of the brain, and already provides a first functional perspective over the whole brain with its separation of neuromeric basal and alar centers. Evolutionary encephalization clearly results mainly from localized differential growth of some *alar subdomains* at specific neuromeres (case of the eyes, the telencephalic vesicles and the cerebellum; Figure 1 and Figure 2C)

The neuroepithelium of each FMU differentiates secondarily via neurogenesis and gliogenesis into ventricular, mantle, and marginal zones stacked parallel to the ventricular and pial surfaces [12]. The ventricular zone contains the somata of the proliferating subpopulation of ventricular progenitor cells. It presents a variable thickness during development as its activity first increases to a maximum and then decreases, finishing as the monocellular ependymal lining of the adult ventricle (still capable of regeneration after a lesion). The mature mantle becomes the thickest part of each FMU and contains locally born neurons (less those that migrate away) plus immigrated neurons and glia cells. Later, after axogenesis (axonal outgrowth) advances and axonal navigation and fasciculation occur, the mantle zone displays diverse more or less organized cellular or fiber-rich strata (the large initial cell masses are known as pronuclei; [12]; they may mature as nuclear primordia, cell layers or reticular formations]). At least periventricular, intermediate, and superficial neuronal strata are distinguished in most places of the mature brain in a pattern characteristic of each FMU. The superficial neuronal stratum, or the whole mantle zone, sometimes forms a multi-laminar cortical structure. This stratified differentiation of the mantle zone is one of the mechanisms that causes additional regionalization (80 FMUs × 3 main neuronal strata = 240 pronuclei). Each pronucleus may later subdivide into several smaller nuclei or layers [12].

Independently of such histogenetic details, each FMU continues registering molecular signaling processes leading to subdivisional patterning (partitioning) of the FMU area, generating a hierarchy of different so-called ‘microzones’ or ‘secondary, tertiary, … to terminal FMUs’ along the dorsoventral and anteroposterior dimensions [12,19]. Each of the resulting subareas behaves as a mini-unit that may use its unique molecular profile to produce one specific cell population, or, sequentially, various specific types of neurons. This final regionalization process is the one that importantly elevates the number of recognizable small brain parts *represented by single cell types* to the highest numbers detected recently ([54]; 5500 distinct cell populations per brain half). Each primary FMU is therefore a stratified composite of anteroposterior and dorsoventral microzones or terminal FMUs that results in production of many unique neuronal cell types (some molecularly differential glial cell types are also possible). Iindividual postmitotic neuronal types may move variously within the primary FMU mantle, becoming roughly intermixed with other cell types, or forming somewhere a compact assembly of a single type. Eventually, groupings of several cell types may emerge that aggregate as characteristic layers of corticoid assemblies (this is for example how the layered spinal dorsal horn is produced, with diverse neuron types each originated in different microzonal domains).

The existence of numerous examples of tangential migration in the developing brain implies that some neuronal populations originated at one specific FMU may spread out via active tangential migration (with or without preceding axonal outgrowth) to another FMU or several other FMUs (close or distant). The stabilized immigrated neurons participate within the host FMU in specific local circuitry according to the molecularly regulated adhesive selectivity and synaptogenetic or trophic potency of each neuronal type (sometimes depending on local functional activity and emergence of biological advantages of their incorporation). Defining completely the *functional role* of a FMU is thus a complex task, needing to distinguish its intrinsic neuronal constituents (locally produced neuron types) from those originated there but migrated elsewhere (emigrated neurons possibly participating in other functions in different FMUs) and, finally, those neurons each FMU may receive as immigrants from other FMUs (collaborative immigrant neurons). About 25% of cortical neurons are immigrated subpallial inhibitory neurons, and 90% or more of the neurons in the olfactory bulb are similarly immigrated from the subpallium. 100% of the pontine and inferior olivary neurons are migrated from the rhombic lip (dorsalmost alar plate in the same or different neuromeres). Most thalamic inhibitory interneurons are immigrated (and their number increases enormously in primates). This final complex spatial production pattern, diversification and functional intermixing of neuronal types is what explains that the 80 *primary FMUs* produce finally over 5500 neuron types throughout the brain (an average of 6–7 neuron types or more in each primary FMU).

The combined functional properties of His’s longitudinal zones (motor versus sensory functions) and of the neuromeres (e.g., reflex or other dorsoventral modular integration of sensorimotor response mechanisms at different segmental levels) would accordingly be distributed with different degrees of specificity over the ‘fundamental morphological units’ or FMUs (19,12]). Note each FMU is initially molecularly unique, having a peculiar *molecular profile*, with a number of shared genetic determinants and a set of *non-shared differential genes* (normally several hundreds of activated or repressed genes, with their eventual downstream cascades). The FMUs may thus express both some common histogenetic properties with adjacent units—e.g., those useful for forming a plurisegmental column, attracting across several neuromeres, for instance, a particular subtype of afferent nerve fiber. In parallel to this there emerge also some singular properties differentiating the FMU’s components from others belonging to other neuromeres. The finer levels of neuromeric regionalization occur in the so-called terminal FMUs or *microzones*. Their occurrence reveals that the initial molecularly unitary FMU later becomes subdivided in its particular pattern of microzones (the latter are also specified with molecularly unique combinations of hundreds of genes; see a well studied pretectal example in [12]).

More complex pathways and functions would be explained via selective connections, and higher development and diversification of collaborating cell populations over evolutionary time, with emergence and new functional role adoption of centers such as the cerebellum and the telencephalic cortex, subpallium, septum, and amygdala (see [55,56]). The theory of ‘fundamental morphologic units’ implying differentially fated neuromere-derived subareas of the embryonic neural tube wall already allows to explain many detailed aspects of adult structure and function within columnar macrodomains on the basis of the selective properties manifested by modular columnar subunits belonging to different neuromeres and neuronal subpopulations residing within the diversely stratified terminal FMU derivatives of each neuromeric submodule. The early simpler neurohistologic notions of the mid-fifties (the synthetic neuromeric model represented in Figure 2B) could not be developed in detail until advances occurred in molecular biology and genetics, and resulting developmental patterning studies, which could underpin more precisely *how* the terminal FMUs get their respective fates specified, and subsequently develop their different molecular profiles step by step into different cellular and histic fates (nuclei, layers, etc.…; see Figure 2, Figure 3 and Figure 4). Detailed functional analysis will need to wait until the classic grossly-defined columnar functions are redefined in terms of the FMUs, their internal specific circuitry, and their external connective properties. The structural and functional plasticity associated to adaptation and learning shown by the mature brain is another significant aspect that needs to e considered in the modern scenario of FMUs.

The earliest modern responses to these questions arose in the early nineties, once developmental genes started to be mapped, first in *Drosophila* (review in [57]) and later in vertebrate models such as the mouse (or zebrafish, frog, chick, etc.…; e.g., [58], or [59,60]). 

### 2.6. The Birth of the Molecular Prosomeric Model

A new molecularly-based synthetic neuromeric brain model known as the ‘prosomeric model’ was produced in the early nineties. This was essentially a variously updated and technically perfected molecular development of the notions previously advanced by the ‘synthetic neuromeric models’ of the 20th century twenties-eighties, with a shared basis on the intersection of the original old ‘early neuromeric model’ and ‘His’s model’ of the 19th century (Figure 1 and Figure 2C; note that in recent years we have realized within the prosomeric model that the molecularly defined forebrain includes the midbrain, while the hindbrain and spinal cord can be distinguished as separate tagmata diverse in primary molecular constitution; see [61] for evo-devo underpinning of this conclusion. The modern ‘prosomeric model’ was developed in progressive stages in continuing collaboration by the Puelles lab in Murcia, Spain and the Rubenstein lab in San Francisco, California, in work starting in 1991. A preliminary chick version of such a synthetic molecular model, though still lacking gene data [51] was based on whole-mount histochemical mappings of acetylcholinesterase-positive young neurons in the developing chick brain (performed by the Puelles lab between 1977 and 1986). The results obtained on the heterochronic topography of neurogenetically advanced or retarded neural wall areas were inconsistent with the reported homogeneity of the hypothetic diencephalic columns supposedly formed across effaced interneuromeric limits, as described by Kuhlenbeck [13,14]. These mappings of AChE-positive young neurons revealed instead the existence of cryptic persistent transverse neuromeric boundaries across the purportedly homogeneous columns. It was eventually shown experimentally that such neurogenetic areal limits coincide with limits of neuromeric gene expression patterns (Figure 3 and Figure 4). Amat et al. [41] recently reexamined in retrospect (after 35 years) these and other AChE neurogenetic chick results (contrasting chick data with comparable lizard and rat embryonic data). The initial conclusions of 1987 were partly updated in 2022 in the light of more recent knowledge. 

Rubenstein and Puelles started to collaborate after a casual encounter at the 1991 Anaheim meeting of the American Society of Neuroscience, which led to discussing the advantages of alternative brain models for morphologically meaningful interpretation of developmental gene expression patterns. We started to work on the neuromeric interpretation of diencephalic, hypothalamic and telencephalic developmental gene patterns in mouse embryos, leading subsequently to several high impact publications presenting the mouse ‘prosomeric model’ (e.g., [52,62,63]; the model was subsequently stepvise revised and updated in [18,20,54,64]; Puelles provided a personal account of how this model emerged [5]).

The model was thus updated several times in recent years, largely as a result of improvements in the causal (molecular or experimental) definition of the longitudinal dimension of the forebrain, as manifested in molecular and fate-mapping studies of the rostral ends of the floor, basal, alar, and roof plates, as well as of the alar-basal and interneuromeric boundaries. A deep analysis of the hypothalamus [64,65]) added the new concept of a transverse ‘acroterminal area’, conceived as the rostromedian end of the whole forebrain acted upon in anteroposterior patterning by the endodermal prechordal plate at the beginning of gastrulation. This hypothalamic rotralmost subarea -already suggested by [32,33]-connects the rostral end of the floor with the rostral end of the roof plate across the intervening fused rostral ends of the basal and alar plates; review in [18,20]; see also [41]). Causal molecular correlation of the brain’s axial dimension read at different dorsoventral positions introduced more precision than had been possible before in the molecular and anatomic definition of the interneuromeric boundaries (see also [16,20,53,66]). Between 1993 and 2005 experimental fate mapping of neuromeres in the hindbrain and diencephalon was performed in combination with relevant molecular markers and additional gene maps, which allowed the extension of the prosomeric model to hindbrain regions.

Between 2008 and 2013 the prosomeric model (Figure 1 and Figure 2C) was incorporated by L.P. to the reference atlas drawings he developed for the Allen Developing Mouse Brain Atlas (Allen Institute for Brain Science; Seattle, WA, USA; developingmouse.brain-map.org; this is a public database with some 4000 developmental genes mapped across 4 prenatal and 3 postnatal stages plus the adult brain; see [67]). The Developing Mouse Brain Ontology developed by L.P. jointly with this project (see this Ontology under Supplementary Data in the cited webpage; see also [17,68]) elicited over 2500 identifiable embryonic sites in each half of the mouse brain, as was mentioned above. Note this ontology considers not only transverse and longitudinal divisions, but also radial ones (strata in the mantle), a feature of the more advanced stages of histogenetic development. The Allen Developing Mouse Brain Atlas has become since a substantial aid for developmental brain studies around the globe. Recently, an MRI- and CCF-based 3D version of the Allen Developing Mouse Brain Atlas has been developed [69].

In essence, the ‘prosomeric model’ holds to the synthetic view that the primary developmental units (fate units) of the brain primordium are represented by the checkerboard-like intersections between the anteroposterior series of transverse neuromeres and the dorsoventral series of His’s longitudinal zones, defining what may be indistinctly called primary migration areas, radial histogenetic areas, or fundamental morphogenetic units (FMUs; [5,12,18,19,20]). This means that each neuromere produces its own set of primary floor, basal, alar, and roof plate areas, which may show either invariant (shared) or differential characteristics when corresponding portions of neighboring neuromeres are compared (Figure 3, Figure 4 and Figure 5). Each final FMU subarea acquires a characteristic molecular profile via anteroposterior and dorsoventral molecular patterning of the progenitor neuroepithelial cells (corroborated in many cases by published gene mappings and recently also by single-cell transcriptomic technology; [54]). Importantly, each of the primary or fundamental structural units is further regionalized secondarily into a number of smaller *terminal FMU areas or microzones*. The latter are neuroepithelial partitions within the primary FMUs that represent progenitor areas for *specific neuronal populations* (i.e., they represent true minimal units of brain structure). Each of them shows likewise a differential molecular specification underlying the typological diversity of the neurons produced. Once the early molecular patterning phase occurring at the level of neural progenitors is settled (activating given genes and repressing others in a combinatorial unique positional code), the resulting molecular profiles start to operate differentially at each locus (by activating different gene cascades, etc.; see [59,60]). Networked downstream final genes produce synthesis of multiple coded proteins, and this leads to early particular phenomena of patterned proliferative growth and cellular differentiation (neurogenesis, gliogenesis), in the ensuing developmental process. The latter effects are expanded by various other special molecular and histologic differentiative mechanisms occurring in different FMUs, that jointly lead to the structural morphogenesis of each particular site (bigger or smaller) of the adult brain, with permanent conservation of the neuromeric complexes and their transverse limits (Figure 3, Figure 4 and Figure 5).

How many unitary organogenetic parts do we have then in the brain, according to this model? Well, we start early on with the single neural plate, but this soon starts to show incipient patterning of its dorsoventral zones (4 zones) and anteroposterior neuromeric subdivisions (20 neuromeres in the forebrain and hindbrain, plus the acroterminal area, and the likewise segmented but species-variable spinal cord). This intersection generates a minimum of 4 × 20 = 80 structural *primary* FMUs (note we are leaving out the spinal cord neuromeres; there are over 30 of them in the human brain). We expect a substantially higher number of definitive terminal FMUs or microzones as well as of individual mantle strata (pronuclei and nuclei), or of cortical areas and layers, than have been described so far. We ignore still whether all primary FMUs subdivide into a similar pattern of terminal FMUs or microzones (though variable partitions seem to occur according to present partial data). At spinal and hindbrain levels, where existing studies are more abundant and precise, there seem to be about 10–11 *dorsoventral microzones* per neuromere. Generalizing tentatively this pattern to the forebrain and hindbrain we may have up to 20 neuromeres × 10 microzones = 200 different dorsoventral progenitor microzones in the brain. Each of them is going to produce via molecularly modulated differential histogenesis and morphogenesis a separate and molecularly (typologically) distinct neuronal population complex of the final brain anatomy (whose total number of different cell types in half a mouse brain ascends presently to some 5300–5500 [54]. We know in addition that some neuromeric alar domains also become subdivided into 3–4 *anteroposterior microzones* (see [70,71]), but we ignore still how general this pattern is.

In any case, we cannot have a complete brain without the whole set of developing units performing their individual morphogenetic roles. The different neuronal types produced in the microzones may combine into adult nuclei or layered cortical structures (both of which usually have mixed subpopulations (physiologists classify them into functional types according to specific differential properties and morphologic phenotypes). Varied adhesive properties modulated by the respective molecular profiles guide the axons of the projection neurons as well as those of interneurons through their respective navigation patterns, to reach more or less distant specific targets (final synaptic contacts seem to be permanently optimized according to functional efficiency and biological, personal and social needs, e.g., in creating memory traces or perfected hypocomplex networked structures). The rough calculations above suggest that the sum of all microzones probably produces several thousand neuronal types, as has been corroborated transcriptomically [54], part of which recombine topographically via tangential migrations with diverse functional consequences. Individual FMUs and microzones will each grow diversely, so that some of them may remain rather small (particularly if they lose many neurons by migration), whereas others will grow more, and still others may grow enormously (and probably produce additional tertiary or quaternary subdivisions, as happens in the cerebral cortex).

## 3. Functions

### 3.1. General Functional Implications of the Prosomeric Brain Model

I’ll address here only non-conventional functional properties arising from the postulated neuromeric subdivision of the brain of vertebrates. These properties have been only exceptionally considered in the physiological literature, but interest in them is increasing in recent years.

The 20 cerebral neuromeres contemplated in the prosomeric model provide numerous functional possibilities due to both their *shared structural and cell typological aspects* joined to their additional *unique individual differential aspects*. The following analysis applies as well to the spinal cord, which was always thought to have a metameric segmental structure, due to the obvious presence of a spinal nerve pair in each neuromeric unit. Clinicians have been aware for a long time of the segmental significance of objective dermatome and myotome bands in the body of their patients, serving to determine the spinal level of their injuries.

Precisely, one of the shared functional aspects of nerves, both afferent and efferent, is how they distribute their peripheral terminals in the body in a strict somatotopic pattern, forming topologically transverse bands called dermatomes in the skin and myotomes in the muscles. Interestingly, successive adjacent nerves always mix their terminals to a degree. This feature is held to aid the central tying together of the data vehiculated by multiple nerves. When we hold a glass of water in our hand, we simultaneously obtain hundreds of different somatosensory signals running through diverse nerves. Their collective input is collected and analyzed centrally, where our unified perception of the object we are handling arises.

Curiously, rather than targeting a point-like locus in a central map of the body, most afferent nerve fibers bringing in information from a particular sensory field distribute it to *an elongated receptive field of secondary neurons distributed over a number of neuromeric units*. Indeed, in the spinal cord and brainstem, afferent sensory nerve roots systematically bifurcate into ascending and descending branches as they enter the brain. The spinal descending branches connect with postsynaptic neurons in two segments below their level of entrance, whereas the ascending ones do the same along four overlying segments. Each segment accordingly receives input from 6 nerves, and thus information from 6 dermatome loci. The next segments above or below it will share 5 of these inputs; each will differ only by one input among 6, and this pattern is reproduced segment by segment along the whole representation of the body. The slight mixing of peripheral origins of the inputs that the terminals of one nerve obtain in a dermatome with those spread out in the immediate dermatomes is multiplied by 6 centrally. This systematic organization clearly forces the existence of secondary computing circuitry reproduced (shared) in all segments for detecting what sort of stationary or moving stimulus is impacting *their part of the body*.

As we see, the neighboring neuromeric units involved in any perceptive task share most of the data they get from 6 successive nerves (5/6), and differ uniquely with adjacent neuromeric units by one differential nerve input added above or below within the set of its 6 afferent nerves. For many purposes, any segment probably can do most of what its neighbors do, and can thus eventually substitute for them with little loss of information. In the dorsal horn module of each spinal segment, a modular part of the larger *plurisegmental columnar* dorsal horn structure that receives the trunk and limb somatosensory signals, the 6 sets of nerve terminals are represented in somatotopic order along a mediolateral distribution. Each modular segment thus has its own small partial map of the somatic dermatomal periphery attended by the 6 potentially *similar or different inputs* it receives. The dorsal horn also disposes of abundant interneurons and both the intrasegmental and intersegmental circuitry needed for cross-correlating the different inputs across the series of partial maps and filtering out a conclusive stimulation pattern for the projection neurons which will relay signals to the next, more integrative, analytic part of the central nervous system, eventually the thalamus or the dorsal column nuclei, where a cohesive central representation of the entire body surface can be obtained in an analogous summatory and internally cross-correlated way.

Such patterns of partial plurisegmental central analysis of peripheral somatosensory nerve inputs occurs as well in the brainstem via the trigeminal, facial, glossopharyngeal and vagus afferent fibers. The principal somatosensory nerve, the trigeminus, displays a small ascending root distributing to its main sensory nucleus, restricted to rhombomere 2 (or perhaps r2 and r3), and a long descending root that distributes terminals to the descending trigeminal sensory column along the rest of the more caudal brainstem (pontine, retropontine and medullary levels; rhombomeres 3–11). The descending trigeminal tract additionally penetrates into the cervical spinal cord, reaching apparently down to its 6th segment, intermixing accordingly face data with neck data. Each dot-like signal originating from any place in the face, eyes, or mouth is thus *copied to redundant circuitry* in 16 successive segments (10 rhombencephalic, 6 spinal) across brainstem and adjoining spinal cord! The three peripheral branches of the trigeminal nerve (ophthalmic, maxilar, mandibular) are each represented by terminals in dorsoventral somatotopic order throughout this extent. The corresponding somatotopic signals are thus computed redundantly in all these segmental units. In this case we know that the different rhombomeres have each a differential molecular profile due to early anteroposterior patterning (*Hox* genes, gradients, etc.…). This causes that the neuronal types produced at each unit may be at least in part qualitatively different, so that *the modular analytic computers that receive the 16 copies of trigeminal sensory signals each are subtly different*! Each module may attend to different variables or combinations of variables of the signals received, and probably generates via unique aspects of intrinsic microcircuitry distinct characteristic output signals capable of targeting different postsynaptic sites in the thalamus or elsewhere. The richness of varied analysis thus made possible by a plurineuromeric structure of the primary sensory inputs stands out.

Less information is available about the mode of distribution of visceral afferent signals, but the solitary tract is also composed of primary ascending and descending afferent fibers from several cranial nerves (facial, glossopharyngeal and vagus) and we know that the column of the solitary tract is also a plurisegmental entity (distributed across rhombomeres 7–11) which has specialized functions in some of its modules (e.g., gustatory analysis at its rostralmost module).

The cochlear sensory column extends over rhombomeres 2–6. We know that auditory afferent fibers enter at r4 and bifurcate into ascending and descending branches that synapse in a tonotopic redundant pattern over all these segmental units.

It is also known that different types of neurons are characteristic of each of the sensory cochlear modules (e.g., bushy, stellate, and octopus cells), and each cell type produces a characteristic output signal addressed to a particular hierarchically superior target. The same may be said of the vestibular sensory column, which extends across rhombomeres 2 to 10, with the root at r4. The vestibular afferents also bifurcate into ascending and descending branches and connect in an ordered redundant way with all these modules, where specialized sorts of neurons are contacted, and a variety of functional output pathways are generated [13].

The optic tract distributes retinal output signals of several subtypes (depending on the ganglion cell types that originate them) to retinorecipient centers distributed over hypothalamic, diencephalic and midbrain prosomeres (hp1-2; dp1-3; m1; review in [30]). Some kinds of functionally dedicated retinal output go to dorsoventrally or anteroposteriorly different target neuropiles (e.g., the signals of melano-photoreceptor cells to the hypothalamic suprachiasmatic nucleus–at the hp2 segment- or the olivary pretectal nucleus -dp1). Some other retinal fibers (note all of them are descending, since the eyes and optic chiasm lie at the rostralmost acroterminal domain of the forebrain) distribute terminals to several prosomeres, and establish multiple separate retinotopically organized synaptic fields having differential analysis properties and projection targets (Figure 5). The conventional columnar model continues holding that the sensory optic input enters the forebrain through its floor [29].

The olfactory input is peculiar in many ways. All its primary projections coming from the sensory cells in the nasal olfactory placodes fall into a single synaptic target which forms a specialized part (the olfactory bulb) of a specialized part (the telencephalon) of a *single* hypothalamo-telencephalic prosomere (hp1). Actually, the whole olfactory pathway into cortical analyzers (signals only reach the thalamus collaterally, in the dp2 prosomere) is also characterized by redundant projection of the same signal to different specialized telencephalic olfactory areas, each of which performs its particular computation and projects to its more or less unique targets, in this single case strictly without recourse to multiple neuromeres.

We have thus seen that developmental serial segmentation in general produces repeated modules with some degree of differential molecular specification (and inner secondary regionalization), together with some degree of sharedness in the connective properties needed to receive in a particular topologic order signals from diverse types of afferent channels vehiculated by sensory nerves. The resulting neuromeric structure is modular and redundant in various ways, but achieves nevertheless, thanks to the accompanying molecular differentiation of variant sorts of neurons in each module, a subtle capacity to compute differently the copied messages received, producing peculiar outputs serving different purposes. A single sensory receiving center as the simpler terminus for all modalities would not be able to extract as much biologically relevant information as the series of variant neuromeric modules. Moreover, the existence of various modules which are partially similar in their analytic properties establishes a reserve for cases of punctual lesion or functional failure of one module, whose essential functions possibly can be saved by an increase in the functionality of its companions.

What about the motor centers? Are they also organized into redundant segmental modular entities? Both cranial and spinal motoneurons have been mapped to aggregates (nuclei) collecting all neurons that innervate specific muscles. Some of these aggregates, particularly in the spinal cord, reportedly extend across several segments. Even though some cranial nerves (e.g., the trigeminus, or the vagus and the hypoglossus) also vehiculate motor axons from motoneurons lying in different neuromeres, generally these target specifically different muscles with characteristic functions (e.g., all trigeminal motoneurons in rhombomere 2 apparently collaborate to close the mouth -the mandibles- with more or less force, using various muscles; in contrast, those sitting in rhombomere 3 apparently serve the easier task of opening the mouth, also with several muscles). In any case, the innervated muscle fibers are aligned across diverse muscles within myotome bands, which is a peripheral segmental pattern.

Most movements imply the activation of multiple muscles, which may change as the task proceeds in a quite dynamic pattern. The important feature is the activation of a particular *proportion* of the fibers of particular muscles (these will automatically alternate activity with other fibers of the same muscle in similar proportion to obtain transient repose; similarly, some muscles partly substitute for others across the same articulation, at least part of the time, when lesions or significant tiredness occur). Often the motoneurons innervating the same muscle are mutually connected via electric junctions; these cause that they discharge in a synchronized way, at least above a certain signaling threshold. Here, again, we have the junction of several actors -now the peripheral effectors, the muscles- that share a generic functional capacity at a particular articulation, but also have particular specialty or variant properties (e.g., various angles of action) that are conveniently used (chosen) when it is needed

Apparently, the putaminal striatum is an important telencephalic center for the selection, dimensional organization, and heterochronic activation of the different muscles involved in an action. Its inhibitory output is vehiculated across the likewise inhibitory pallidum and the excitatory thalamus to the cortical motor pathway (see below). The descending pyramidal tract fibers apparently give out collaterals at different segmental (neuromeric) levels throughout the midbrain, hindbrain and along the whole spinal cord, allowing the cortex to act freely upon the entire keyboard of segmental motoneuron groups.

It is not yet clear precisely how the central nervous system ‘thinks’ about the distribution of the action intended across the segments in which the multiple relevant motoneurons are placed and the myotome units where the final targets, the individual muscle fibers, are found (all this being in equilibrium with the entire acting body and the outside world).

In an initial phase, one presumes that the premotor cortex conceives and selects the action to be performed in a broad context wherein many other alternative actions are conceivable; this generates a sketchy draft of a plan (e.g., you think of ‘going out for a walk’, without counting and planning the steps or corners implied). This sketch apparently passes via corticostriatal projections through the striatum, which selects a more detailed plan involving specific muscular units in a precise dimensional context (are we moving only a finger, a limb, or the entire body?) and ads planned exertion of force by each unit over time, according to apparent cortically envisioned necessities and helped by recorded memories of previous similar actions (non-practiced actions are more difficult to plan and are uncertain of success). The striatal updated action plan passes to the thalamus translated into a set of disinhibition signals via the striato-pallidal pathway, modulated by parallel excitatory signals from the subthalamic nucleus. The disinhibited thalamic neurons relay excitatory trains of signals that modulate the current output of the primary motor cortex, itself pre-stimulated by the previous premotor plan. The thalamus also relays relevant vestibular, visual, auditory, tactile and propioceptive sensory inputs serving to orient the action in the world and the body, as well as crucial standing cerebellar corrections of the precision with which the plan in course achieves its objective (this is a feedback mechanism activated by collateral copies of the intended movement that reach the cerebellum via the pons apart of relevant parallel propioceptive, visual and vestibular cerebellar inputs; the cerebellum compares current state dynamics with the motor plan and generates error correcting signals for the thalamus to send to the cortex). Associative cortical networks integrate the different sets of data transmitted about the specific action being performed in the changing context of other concurrent mental contents.

All these connections needed to organize any simple movement use selected groups of neurons sitting at different neuromeric modules, which thus collaborate on demand more or less, depending of the case. Descending motor orders generally reach massively a series of reticular neurons distributed across most rhombomeres and only a selection of these units (those which result more strongly activated perhaps by the contingent coincidence of diverse concurrent inputs) release in their turn relevant reticulospinal pathways doing the same in the spinal cord (redundant multisegmental signaling to a surplus of segmental units). Spinal premotor interneurons do the final local refined filtering and selection of motor signals, thus obtaining the final motor instruction for the motoneurons. There is only minimal direct spinal motoneuronal control by cortical pyramids.

In a way, therefore, the availability of multiple downstream neuromeric units, all of which have their own sensory inputs and capacity to organize details of reflex or voluntary motoneuronal output in mutual interaction, eliminates the need for the cortex to compute absolutely all details of intended actions. This system also has the potential to learn over time habitual action mechanisms with many subtle variants, which later can be selected from and repeated automatically with modulated force and velocity with surprising success. This bottom capacity for adjustment on the run of any sufficiently practiced motor act is called ‘mechanization of movement’ and is largely dependent on the repetitive, redundant properties of series of neuromeric units that share considerable information among them, but subtly offer alternative variant options among which the higher centers can easily choose. Most of the detailed computing is accordingly pre-prepared. This allows the enormous functional capacity of the cerebral cortex to be dedicated instead to aspects of creation, motivation, combination, and intentional design. A concert pianist at work thinks about the music he wants you to hear, not about his own finger movements. The neuromeres and other sophisticated auxiliary mechanisms are covering that duly mechanized aspect.

### 3.2. The Case of Vision

The sense of vision is peculiar in that its receptive organ, the neural retina is itself an evaginated part of the wall of the neural tube. Evolutionarily and ontogenetically, the retina starts as a single median patch at the rostral end of the neural plate. Subsequent action of signals originated from the median prechordal plate inhibits the retinal fate at the midline and leaves bilateral primordia that evaginate producing the optic nerves and eye vesicles, a central part of which differentiates as the neural retina. This evaginated patch of the neural wall produces a layered neuronal structure that analyzes light-mediated visual input that reaches the deep retinal photoreceptors. The retinal output is emitted along the optic nerve, optic chiasma and central optic tract (divided into a principal lateral part and an accessory basal tract; Figure 5) by the diverse types of retinal ganglion cells. Thanks to neuromeric specializations, a general function, such as ‘vision’ is decomposed into many partial functions occurring in parallel, but mutually coordinated, in different parts of the brain. Neuroscience books often comment about the two cortical visual ‘pathways’, namely those computing on one hand ‘what’ you see, and on the other hand, ‘where/how’ you see it, to which can be added the ‘color’ cortical pathway. These complex cortical processes do not have much to do with neuromeres, but many other underlying visual mechanisms occur at subcortical neuromeric level. Among the seven forebrain prosomeres (2 in the hypothalamus, 3 in the diencephalon -these include respectively the prethalamic, thalamic, and pretectal regions- and 2 in the midbrain), six of them contain visual centers (in most cases several centers each, to a total of some 20 subcortical visual centers; [30]; Figure 5). Textbooks usually refer only to a handful of selected subcortical centers involved in visual function, and never cover the full variety of retinorecioient centers. The lateral geniculate and lateral posterior thalamic nuclei (specializations of prosomere dp2; LG, LPc in Figure 5) are cited obligatorily, because they crucially relay retinal data about both ‘what’ and ‘where’ to primary and associative visual cortex. If you lose these data, you are blind.
Figure 5Graphic lateral projection mapping of the set of retinorecipient forebrain centers over hypothalamus, diencephalon and midbrain, according to the prosomeric model and topographic data from the Paxinos and Franklin Mouse Brain Atlas [72] (image extracted from [30]). This lateral view map shows the course of the optic fibers in yellow, the alar–basal boundary in red, and thin blue lines represent the interprosomeric boundaries (the prosomeres are identified at the top –m1, m2, isth- or at the bottom of the map –dp1-dp3, hp1, hp2). The diverse visual centers of each prosomere are given the same background color, alternating green with red, to better see the groupings and multiple projections in each neuromere. In most cases they are in the alar plate, but sometimes also in the basal plate (medial terminal nucleus shown as a dark blue mass tagged as MT). There is no observable columnar pattern at all (columns should be parallel to the optic tract). The noncolored structures do not receive retinal afferents (but the mesodiencephalic substantia nigra, SNC, was exceptionally marked in grey, for contrast).
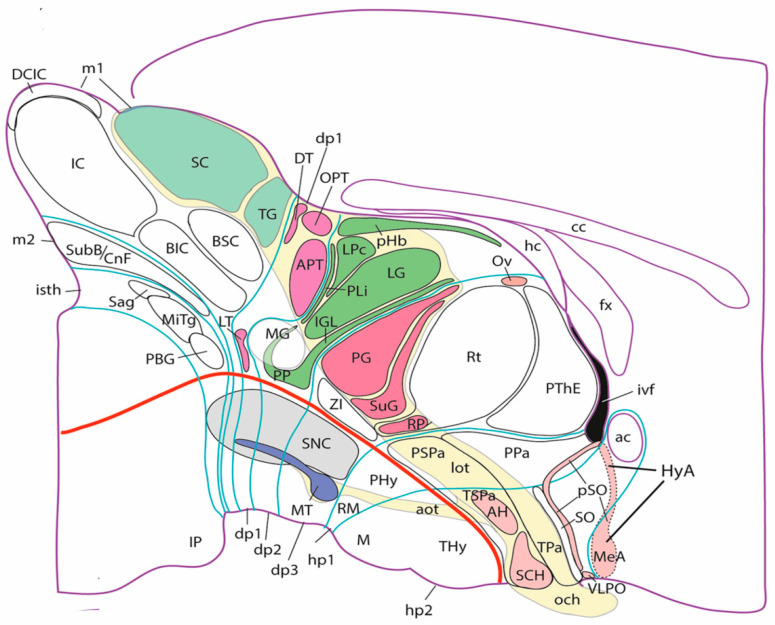



Another frequently cited subcortical visual center is the midbrain superior colliculus (derivative of the rostral m1 prosomere; SC in Figure 5). This large center maps the binocular retinal visual field with respect to your head and shoulders and your auditory field and computes therein ‘where and how you should look’ (i.e., which binocular eye movements are needed to jump to a new place of interest in or outside of the vision field). This can be done under voluntary cortical control (you look where you decide), or partly automatically, responding by reflex to ‘novelty’ signals, that is, to mapped sorpresive visual, auditory or tactile stimuli. Collicular output coordinates indirectly, via dedicated preoculomotor reticular cells in the dp1 prosomere or in rhombomere 5, the degree of activation of each of the oculomotor neurons connected via three cranial nerves to six muscles for each eyeball. These oculomotor neurons also receive parallel vestibular and cerebellar input (via other rhombomeres), which independently adjust your eyes to the accidental movements of your head (for instance in a ship moved by the sea, or riding a horse). If everything works, both your eyes jump rapidly (head movement corrected to eschew apparent movement of the world) to the new mapped points of space that interest us (or at least our superior colliculus), frequently simply because nothing was expected to occur there. We actually ‘explore’ constantly visual and auditory space for such new stimuli of interest. It’s an activity that saves your life in the wild and also in the midst of heavy traffic. You need a good handful of diencephalic, midbrain, and rhombencephalic neuromeres to perform this apparently modest visuomotor jumping function.

Many times, it is not enough to move the eyes in their sockets, and we have to move as well the head on our neck, or gyrate our entire torso to fix the meaningful target (e.g., a dog growls behind you; better have a look). This of course requires descending collicular pathways targeting other oculomotor neurons sitting in the brainstem (rhombomeres 0 and 5), or within cervical and thoracic spinal neuromeres. The superior colliculus also connects retinotopically in ascending direction with many of the other diencephalic visual centers in dp1-dp3, apparently contributing its mapped computation of ‘where’ in space the things we see are, and how they move relative to us, irrespective whether we look at them or not (via specialized sorts of neurons). Thus, such an apparently easy function like ‘looking’ in voluntary or automatic modes needs a very complex point-to-point interconnected multiplicity of collaborative visual neuromeric centers with associated integration of diverse other sensorial inputs from other neuromeres (vestibular, propioceptive, tactile, auditory), finishing in complex bilateral motor control of two independent eyes so that their lines of sight converge unfailingly (and rapidly, without searching) at the same new point in space. This selects the signal to be studied by the cortex and simultaneously resolves many underlying urgent practical problems not worth cortical attention, which needs time to think. All the axonal pathways involved in this complex network refer their precise growth and synaptic targeting to neuromeric molecular coordinates, neuromeric differential adhesivity properties, and recognition of relative positions of targeted neurons within their respective neuromeric spatial frameworks. You probably are starting to understand now why the columnar theory of the brain does not provide sufficient explanation of brain functions, and we need not only a basic neuromeric perspective but also intraneuromeric hypercomplex microzonal regionalization and single-cell properties to make a dent on the cognitive and practical task ahead of us.

The pretectal region (in dp1; Figure 5) contains apart from preoculomotor reticular cells a dorsal retinorecipient visual center, the olivary pretectal nucleus (OPT in Figure 5), which is involved in the adjustment of the diameter of our eye pupil to the level of light that incides on the eye (the pupil closes automatically proportionately to the level of incident light). This apparently is a simple sensorimotor reflex task (we regulate the pupil with a constrictor muscle controlled via the parasympathetic ciliary ganglion lying close to the eye) but it also involves at least coordination of the three diencephalic neuromeres, each of which has a center (different in terms of neuronal types) receiving via the optic tract a copy of the ‘how much light’ signal from the retina, so that the resulting reflex output via a single nerve is either summatory, or can be independently triggered from three different retinorecipient sites, or is triggered in three diverse functional situations (e.g., you suddenly dilate your pupils when you are insulted or are otherwise attacked, irrespective of the level of illumination, because your parasympathetic tone decreases as the sympathetic tone increases suddenly; the sympathetic tone is regulated from the thoracic spinal cord). You need analysis by 3 diencephalic neuromeres, a midbrain neuromere for initial output to the peripheral ganglion that controls the muscle, and autonomic parasympathetic and sympathetic output ganglia for understanding this simple circuit.

Daily time cycles are measured by a central neural clock, the suprachiasmatic hypothalamic nucleus (SCH in Figure 5). This receives separately information from the retina about the amount of ambient light to which we are exposed (maximal at noon). This signal is translated and projected widely to the forebrain and hindbrain neuromeres, with widespread effects on all metabolic, temperature-control, hormonal, activity, and sleep/awakeness cycles that configure our body state in ways that change along the day.

I have explained briefly the 5 best known subcortical visual centers, whose specialized functions related to retinal signals commonly involve interactions with specialized vision-related neurons in various neuromeres. There are however actually about 20 such subcortical visual centers (review in [30]; Figure 5), most of which are still uncharacterized about which functions they perform and which are the relevant interneuromeric circuitries involved. There are particular visual functions which have been partly studied physiologically, but we still ignore which is the contribution of particular neuromeres to them. That is the case of ‘accommodation’, that is, the focusing of our perceived sight according to the relative distance from us of the object of interest. Unfocused objects look blurry or duplicated. Important knowledge has been gained about the cortical mechanism that examines binocular images and *identifies quantitatively* any apparently duplicated mental images implying an unfocused single object (rather than resulting from real double objects). This diagnosis can only be performed in the visual cortex. It is assumed that the cortex consequently sends a descending correcting signal to some subcortical visual center in the diencephalon and/or midbrain (5 neuromeres in those two regions) and this center would translate the cortical correcting input and send a corresponding motor output perhaps via reticular cells to the oculomotor motoneurons involved in ocular convergence (innervating the medial rectus muscle of both sides). Their activation brings the visual focus closer to us (or obtains via inhibitory interneurons a relaxation of the tonicity of the same mechanism for focusing more distant objects). There is no insight yet on where this subcortical accommodation mechanism lies, but I would not be surprised if it also involves collaborative modular neuromeres.

Any other major neural function can be similarly decomposed into a set of different neuromeric collaborative networks that interact to contribute partial functions that collectively achieve and apparently ‘make easy’ the performance of quite complex neural tasks [73]. There is current physiological research going on on neuromeric centers involved in respiratory rhythms, cardiac rhythms, vocalization, temperature control, sleep/awakeness cycles, hunger/satiety cycles, or vestibular, cochlear, and trigeminal neuromeric functional modules (e.g., [74] see also [19]). The mechanism involving ‘place cells’, ‘border cells’, ‘head orientation cells’ and ‘spatial reticle cells’ in the hippocampal system apparently is underpinned by various neuromere-related propioceptive, vestibular, visual, hypothalamic (mamillary) and thalamic subcortical subsystems (e.g., [75,76]). We know that various central modulating neuronal systems, such as the dopaminergic, noradrenergic and serotonergic systems, plus the still largely functionally mysterious interpeduncular nucleus, are organized neuromerically (see, e.g., [77,78,79]).

## 4. Conclusions

In the beginnings of neuromeric embryological studies none of the contemporaneous physiological questions could be addressed for lack of sufficient causal understanding about brain differentiation and morphogenesis. This led to attacks on the interest of neuromeric phenomena described by morphologists, because no general function could be ascribed at the time to these developmental units, wrongly thought, moreover, to be transient. Now we know that neuromeres do not disappear, since they become instead fundamental primordia (FMUs) of entire adult brain portions, as has been tested by chirurgical and transgenic molecular fate-mapping (Figure 3 and Figure 4). Such embryonic regions obtain early on a differential molecular specification on top of fundamental shared metameric structural aspects (i.e., a sensory alar plate and a motor basal plate; note this makes each neuromere a minibrain). These initial conditions end up establishing a crucial role for neuromeres and their inner microzonal subdivisions. The latter each generates one or several functionally specific types of neurons that interacts synaptically with several other types, often establishing dynamic networks. As development advances, the specialized neurons enter into complexes (nuclei, reticular zones, or cortical layers or columns). These produce complex computing structures and associated circuitries intrinsic to each neuromere that underpin specific sensorimotor reflex properties, as well as more or less extended longitudinally projecting axons that interconnect metamerically via multiple segmental collaterals particular sensory and motor plurineuromeric columns. The latter always have a modular segmental skeleton related to partial specialized functions. Columns, where they exist, are never neuronally homogeneous, as was mistakenly concluded by Kuhlenbeck [34,35] within columnar theory, because columns are composed of a set of neuromeric modules participating in a common computation (for instance, the trigeminal column integrates 16 neuromeres in the computation of rich and varied facial somatosensory feelings; see also Figure 5 for visual computations). The classic columnar ‘thalamus’ was not a column, but actually referred to badly studied prethalamic, thalamic and pretectal neuromeric modules receiving direct visual input (see Figure 2A,C, showing how what is thought to be longitudinal in one model in fact is transversal in the other, consistently with gene expression underpinnings. The differently colored retinorecipient neuropils in Figure 5, involve as we now know some 20 differential visual microzones serving particular subfunctions of vision via specific plurineuromeric collaboration. The subcortical neuromeric centers of visual subfunctions are connected bidirectionally as needed with the visual cortex functions via the corticopetal parts of the thalamic neuromeric unit.

Modern mechanistic understanding of all major cerebral functions, even those involving the cerebral cortex, the hypothalamus, the thalamus, and the cerebellum, need attention to the subcortical set of neuromeres that provide essential functional support in the realms of sensory analysis, reflex circuitry, evaluation of optimal interlocutors, and prediction of experience, social interaction, or motor organization. The prosomeric model takes into account the smallest structural units emerging during brain development (as microzonal components of neuromeric FMU macrounits). It is becoming an essential guidance model for the mapping and causal analysis of functional circuits and pathways and singular neuronal populations in the brain.

## Figures and Tables

**Figure 1 biomolecules-14-00331-f001:**
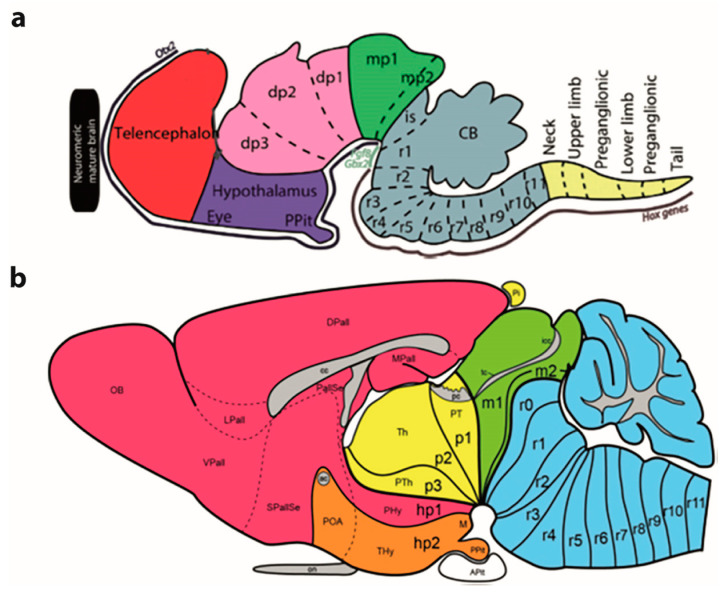
Schemata illustrating embryonic (**a**) and adult (**b**) location and abbreviations for neuromeres within the prosomeric model. (**a**) Updated schema taken from [1], in which the forebrain prosomeric fields (including the secondary prosencephalon [sum of red evaginated telencephalon and violet hypothalamus], diencephalon [pink], and midbrain [green]) appear under the control of the *Otx2* transcription factor. In contrast, prepontine rhombomeres r0 (isthmus) and r1 relate to the area of influence of the secreted FGF8 morphogen and the *Gbx2* transcription factor, while the pontine (r2–r4), retropontine (r5,r6), and medullary (r7–r11) hindbrain rhombomeres plus the spinal cord obey to differential *Hox* gene signals. These markers can be used to identify the different neuromeres in the mature brain [2,3,4]. (**b**) Schema extracted from [5] representing the whole set of prosomeric units in the adult brain (the vermis of the cerebellum belongs to r0—the cerebellar hemispheres to r1). Note the much enlarged evaginated telencephalic development corresponding to hypothalamic prosomere hp1 (red), while the rostral end of the brain corresponds to the rostromedian acroterminal domain **at the front of** hp2 (orange; compare Figure 2C; note the acroterminal optic tract and chiasma as well as the neurohypophysis and adenohypophysis). The hypothalamus is divided rostrocaudally into peduncular hypothalamus (PHy) within hp1 and terminal hypothalamus (THy) within hp2. The axis of the brain bends ventrally under the midbrain (green) at the cephalic flexure, where a number of interneuromeric boundaries converge ventralwards, and also shows a less marked dorsal bending at pontine levels, causing there also some convergence of neuromeric boundaries at the ventricular surface (e.g., r3 and r4).

**Figure 3 biomolecules-14-00331-f003:**
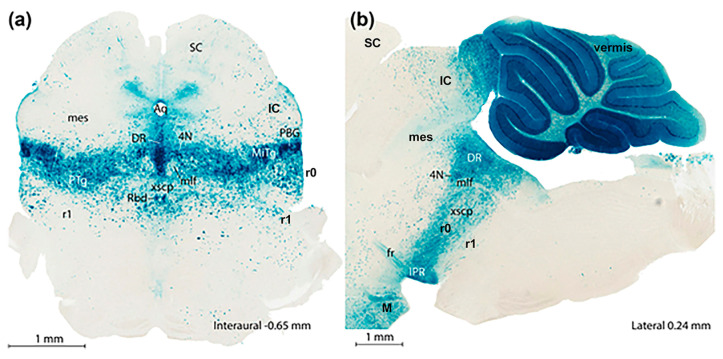
Coronal (**a**) and sagittal (**b**) sections through adult mouse brains reacted with LacZ histochemical reaction to demonstrate the co-expression of the LacZ DNA code inserted transgenically into the *Fgf8* coding sequence (extracted from [1]). This gene is restricted at initial neural tube stages to expression in the isthmic (r0) rhombomere, which is the rostralmost hindbrain neuromeric unit, limiting with the midbrain (mes; represented in (**a**,**b**) by SC and IC). The isthmic neuromere also forms the vermal (median) part of the cerebellum (vermis in (**b**)). The isthmic neuromere (r0), delimited at early developmental stages by *Fgf8* gene expression, is visualized in the adult brain as a transverse blue wedge of LacZ reaction product, including the cerebellar vermis. The unit clearly persists as its adult derivative, still forming a transverse rostral domain of the hindbrain that limits rostrally with the midbrain and caudally with rhombomere 1 (r1). Both neighboring domains show some dispersed blue cells that are interpreted as isthmic-derived cells that migrated tangentially into the neighboring areas. The blue r0 expression domain forms a radially complete transverse ring of the brain (extending from ventricular surface up to pial surface, as seen in cross-section in (**a**)); dorsoventrally it clearly extends from the cerebellar roof to the floorplate interpeduncular nucleus (IPR in (**b**)). The hypothalamic mamillary body (M) lying rostral to the retroflex tract (fr) also shows blue cells, but these are due to separate intrinsic expression of *Fgf8* at that distant locus.

**Figure 4 biomolecules-14-00331-f004:**
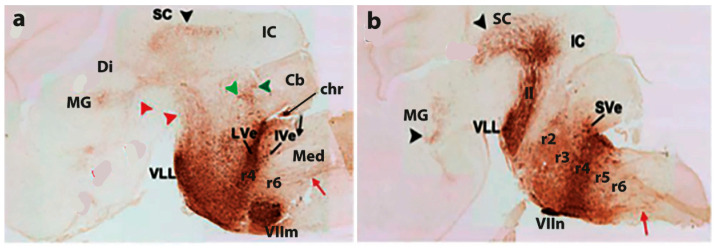
Two parasagittal sections through the same adult mouse brain transgenically labeled with yellow fluorescent protein (YFP) code attached to the *Hoxb1* codon, with brown immunochemical visualization ((**a**,**b**); (**a**) lies medial to (**b**); images extracted from [42]). The *Hoxb1* gene is selectively expressed from very early neural tube stages onwards in the r4 rhombomere. Accordingly, in the adult the YFP labeling indicates neurons born inside r4 and their axonal projections, as well as cells born within r4 and migrated tangentially into other brain loci. Facial motor neurons, for instance, migrate early on into the r6 alar plate (VIIm in (**a**)), and the ventral nucleus of the lateral lemniscus (VLL), present in rhombomere 1 of the propontine hindbrain along the ascending route of the lateral lemniscus (ll) towards the inferior colliculus is also migrated out of r4; it is accompanied by secondary ascending auditory fibers in the lateral lemniscus (VLL,ll in (**a**,**b**)). These fibers originate from the labeled cochlear module of the auditory column that corresponds to r4 (labeled periventricular area above LVe in (**a**)). The labeled lateral lemniscus fibers penetrate both the inferior and superior colliculus (IC,SC), and some of them even follow the brachium of the inferior colliculus (red arrowheads in (**a**)) into the diencephalon, where they reach the medial geniculate nucleus (MG and black arrowhead in (**b**)). A distinct strongly labeled transverse dorsoventral strip represents strictly the adult derivative of r4, identified as well by the presence of the facial nerve root (VIIn in (**b**)). There is also evidence of r4 migrated cells into r3 and r2, as well as into r5 and r6, jointly with projections into the superior, lateral and inferior vestibular nuclei (SVe, LVe, IVe), the cerebellum (Cb; green arrowheads in (**a**)) and the spinal cord (thin red arrows in (**a**,**b**)).

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
