# Peer review of "Functional Implications of the Prosomeric Brain Model"

_biomolecules, 2024, doi:10.3390/biom14030331_

Round 1

Reviewer 1 Report

Comments and Suggestions for Authors

Ιn this exciting essay Prof. Puelles explores current perspectives on brain models, focusing on structural components and organisation in relation to brain axis concepts. The discussion contrasts columnar-longitudinal models with transverse subdivisional neuromeric models, analyzing their functional implications and associated challenges. Special attention is given to the prosomeric model -20 transverse prosomeres - developmental units constructing cerebral parts, forebrain, hindbrain, and spinal cord units. The prosomeric model is characterized by remarkable functional properties arising from its multiplicity and modularity, allowing for the breakdown of brain functions into tasks executed by sets of neuronal elements Additionnaly the neuromeric anteroposterior and dorsoventral positional information influences the order in both neural connections and function. A very useful essay to for every neuroscientist!

Author Response

I appreciate the comments and revised the text accordingly.

Reviewer 2 Report

Comments and Suggestions for Authors

Luis Puelles presents an essay describing the relationship between the anatomical distinct neuromers of the prosomeric model and their functions.

This essay is an important discussion on the functional implications of the model first described by the Puelles and Rubenstein in 1993. 

Introduction

Figure 1 is low in resolution.

Lines 39-42, the sentence in parenthesis should make a new sentence by itself.

Line 44, ‘start’ should be substituted with ‘origin’.

Line 48 of the Figure 1 legend, ‘blue’ should be ‘violet’

Line 72, ¿ should be eliminated.

Lines 87-90, the future clinicians are exposed to a ready to use working frame of the anatomy of the human, and human only, brain without knowing from which models it comes from. Neuroanatomy is taught at medical school because is instrumental only to professional MDs who needs only an adult human brain map ready to use, neurosurgeons for example.

This point could be taken into account here.

Lines 119-120, I will be cautious stating that the brain is the most complex LIVING structural system.

Line 140, 2500 differently named places: places should be better substituted with domains or structures?

Where this number (2500) is taken from? Cite the ontology page or publication.

Line 144, please cite a reference to the Allen project publication and mention the URL of their database. I’ll cite it here than later at pages 634 - 637

Lines 241 – 251, should be made clearer that the adult amphibian brain are erroneously postulated by Herrick to be general landmarks of the longitudinal brain axis.

Line 314, if the Author is referring to the publication by Darwin of “On the origin of species”, the publication date is November 24 1859.

Lines 338-339 the suffix ‘mere’ originates from ancient Greek μέρος (méros), meaning “part”, and implies a ‘repeated partition’.

Line 357, developmental genes should be gene expressed during development.

Figure 2 and figure 3 should be separated by text.

Figure 2 is printed at low resolution.

Lines 509-510, the author should make clear why there are 4 FMUs for each neuromere: roof alar basal and floor subdivisions? And therefore: what is the relationship between FMU and neuromere? Something like: neuromere hp1/roof FMU?

Line 552, I suggest substituting ‘promedium…’ with ‘an average number’ of 6 neuronal types per FMU.

Line 561, eliminate the comma after e.g.

Line 581, histic meaning histological?

Line 591, why the name ‘prosomeric model’? to make a distinction between the old ‘neuromeric model’? The prosomers are the most anterior neuromeres of the developing neural tube, aren’t they?

Line 676, /man should be substituted with human brain.

I would separate lines 1051 to 1082 in a “conclusion” paragraph.

Line 1160, RM should be retromamillary area

Author Response

I revised thoroughly the manuscript following all your comments and suggestions. The Fig.3 was changed to 300 dpi resolution.